# Comparison of ARIMA and LSTM in Predicting Structural Deformation of Tunnels during Operation Period

**Chuangfeng Duan [1,2,3], Min Hu [3,4] and Haozuan Zhang [3,4,*]**

1   School of Mechanics and Engineering Science, Shanghai University, Shanghai 200444, China; dcf@shu.edu.cn
2   Shanghai Urban-Construction Information Technology Co., Ltd., Shanghai 200092, China
3   SHU-SUCG Research Centre for Building Industrialization, Shanghai University, Shanghai 200072, China; minahu@shu.edu.cn
4   SILC Business School, Shanghai University, Shanghai 201800, China
*   Correspondence: zhanghaozuan1999@shu.edu.cn; Tel.: +86-198-1653-2618

**Abstract:** Accurately predicting the structural deformation trend of tunnels during operation is significant to improve the scientificity of tunnel safety maintenance. With the development of data science, structural deformation prediction methods based on time-series data have attracted attention. Auto Regressive Integrated Moving Average model (ARIMA) is a classical statistical analysis model, which is suitable for processing non-stationary time-series data. Long- and Short-Term Memory (LSTM) is a special cyclic neural network that can learn long-term dependent information in time series. Both are widely used in the field of temporal prediction. In view of the lack of time-series prediction in the tunnel deformation field, the body of this paper uses historical data of the Xinjian Road and the Dalian Road tunnel in Shanghai to propose a new way of modeling based on single points and road sections. ARIMA and LSTM models are applied in comprehensive experiments, and the results show that: (1) Both LSTM and ARIMA models have great performance for settlement and convergence deformation. (2) The overall robustness of ARIMA is better than that of LSTM, and it is more adaptable to the datasets. (3) The model prediction performance is closely related to the data quality. ARIMA has more stable performance under the lack of data volume, while LSTM has better performance with high-quality data and higher upper limit.

**Keywords:** tunnel; structural deformation; ARIMA; LSTM; prediction





## 1. Introduction

The shield method is a vital construction method for urban tunnel construction. It is characterized by numerous benefits, such as higher safety standards, faster construction speed, minimal environmental disturbance, and limited disturbance to ground buildings and surrounding soil environments [1]. However, during the long-term operation period of shield method tunnels, multiple factors, which include natural and human triggers, will lead to various problems, such as leakages, cracks, breakages, misalignment, and corrosion [2]. Figure 1 is a photo of tunnel leakage. Tunnel problems have a direct relationship with the structural deformation of shield tunnels, such as longitudinal uneven settlement and transverse convergence deformation [3]. Without continuous monitoring and timely maintenance, once the deformation exceeds a certain limit, it will cause severe safety accidents in the tunnels. To this end, regular testing of tunnel structural deformation during the operation period is necessary, as clearly stated in the Technical Specifications for the Evaluation of Road Tunnel Maintenance Operation in Shanghai [4]. It is essential to anticipate the development trend of tunnel structure deformation, predict risks beforehand, and take early maintenance and repair measures to prevent and mitigate tunnel structural problems, thus ensuring the health and safety of shield tunnels.

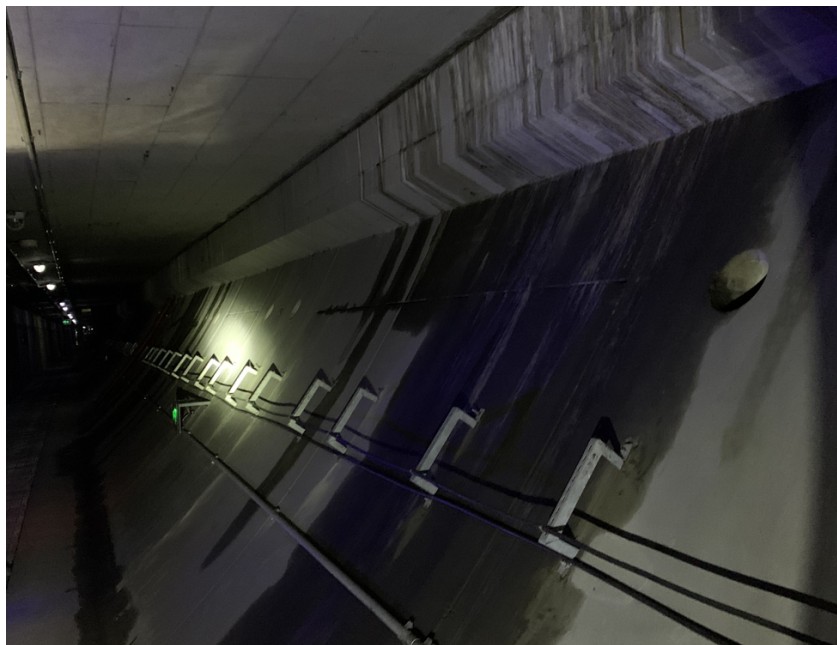

**Figure 1.** Photo of tunnel leakage.

Based on the monitored data and historical information during the operation period of the Xinjian Road and Dalian Road cross-river tunnels in Shanghai, this paper uses ARIMA and LSTM from the perspectives of single point and road section to predict the longitudinal uneven settlement and transverse convergence deformation of tunnel structures and compare the prediction performance of the two methods. The paper explores a more reasonable and scientific structural deformation prediction method during tunnel operation, and provides corresponding suggestions. The following is divided into six parts, including literature review, experiment data, model design, experiment introduction, results and analysis, and summary and prospect.

## 2. Literature Review

At present, researchers mainly have three types of prediction methods for the deformation trend of tunnel structures during the operation period: numerical calculation models, related factor models, and time-series prediction models.

### 2.1. Numerical Calculation Models

The numerical calculation model applied to the prediction of deformation trends in tunnel structures usually combines the principle of the soil consolidation compression and deformation mechanism to simulate the variation of tunnel deformation values over time [5–7]. For example, for a river crossing tunnel, Liu et al. [8] combined the static finite element calculation model and empirical fitting formula to calculate and predict the deformation value, with the number of actions of traffic load and basic parameters of soil layer as important parameters. The methods based on probability statistics have also been applied by some scholars in the field of tunnel deformation [9,10]. Ruan et al. [11] proposed a random variable probability fitting method to determine the early warning value of structural safety of under-water shield tunnels and applied this method to a tunnel structural health monitoring system. Zhang et al. [12] analyzed the deformation behavior of tunnel surrounding rock from a probability perspective by measuring the distribution of data and established a numerical model to predict the deformation trend of the surrounding rock.

However, numerical calculation models rely heavily on the physical parameters and accuracy within the model, making it highly idealized. At the same time, the model parameters are difficult to determine and have poor practicality in various scenarios. If the

parameters of the model are missing or have insufficient accuracy, it will have an impact on the judgment of deformation trends.

*2.2. Correlated Factors Models*

The correlated factor model does not specifically decompose complex tunnel mechanisms, but rather selects appropriate mathematical models, screens appropriate factors, and fits the deformation trend of tunnel structures to achieve the purpose of prediction. The current popular machine learning models [13–16] all belong to the method of using correlation factors for model fitting. Hao et al. [17] used a BP neural network to select seven factors that affect convergence deformation and input them into the neural network to predict the convergence deformation of the Harbin Baojian Road soft soil tunnel. Huang et al. [18] used five environmental factors measured at different times at a settlement monitoring point during the operation period of the Yanshuigou Tunnel as input values for the BP neural network and trained the model with the settlement values at that monitoring point as output.

The limitation of the correlated factor model is that it requires higher data accuracy, and it is difficult to collect numerical values for different factors. The selection of input factors for different models will affect the final model performance and prediction results. The screening of related factors often has a certain degree of subjectivity, making it difficult to determine the input factors for the optimal model. It is also difficult to comprehensively consider multiple factors at the same time. The establishment of factors only targets the current tunnel, making it difficult for the model to predict the deformation of other tunnels.

*2.3. Time-Series Prediction Models*

During the operation period, the deformation monitored data of the shield tunnel are collected at a fixed time interval. There is a certain correlation between the deformation data itself, which contains the deformation development law. The time-series prediction model is a rolling prediction. Compared to numerical calculation models and correlated factor models, it only predicts based on temporal data. Rolling prediction requires low data requirements. Under relatively stable external factors, better prediction results will be achieved. Therefore, researchers [19–21] began to attempt to establish a prediction model based on the time series to predict the trend of tunnel structural deformation. He et al. [22] used the regression method to analyze the relationship between tunnel convergence value and time for the 55 phase transverse convergence deformation data of a section of highway tunnel, and the results show that the tunnel convergence value and time are nonlinear logarithmic. Xie et al. [23] used the ARMA time-series model to model and analyze the measured data of 30 monitoring points of a subway tunnel in Nanjing during an operation period, so as to achieve short-term deformation prediction. Under the feasible conditions of time-series prediction models, it can be found that existing research often predicts based on short-term time spans, while the operating period is a long-term and slow process. To make long-term predictions, the impact of prediction errors and long-term dependence must be considered.

In fact, some more popular time-series prediction methods have already achieved good results in other fields. The ARIMA model is one of the classic time-series prediction and analysis methods, which can effectively handle non-stationary sequences and can therefore be used for long-term prediction [24]. Meanwhile, with the vigorous development of artificial intelligence and machine learning, the Long- and Short-Term Memory network (LSTM), which is improved on the basis of recurrent neural networks, can capture memory dependencies over long time spans and is also a popular method in the field of time-series prediction [25]. Therefore, this article selects these two methods to predict the deformation trend of tunnel structures during the operation period and compares their performance, exploring new ways to predict tunnel deformation.

## 3. Experiment Data

According to the Technical Specifications for the Evaluation of Road tunnel Maintenance Operation in Shanghai [4], the monitoring of shield tunnel structural deformation during operation includes both tunnel settlement and transverse convergence deformation, which is typically conducted through special testing methods for data collection. Figure 2 is a photo of the measuring instruments used for deformation data collection. Figure 3 shows a schematic diagram of the layout of the monitoring points for tunnel cross sections. Settlement monitoring points are arranged in a longitudinal line shape on the top of the tunnel. In some special areas, such as the connection between the working well and the shield section, monitoring points may be added to shorten the distance between them. The monitoring object for transverse convergence deformation is the transverse diameter of the tunnel cross section. Two monitoring points are arranged at intervals on the deformation monitoring section to measure the length of the diameter. The collection cycle for both types of data is generally once every quarter, with a possibility of an increase in monitoring frequency for special cases.

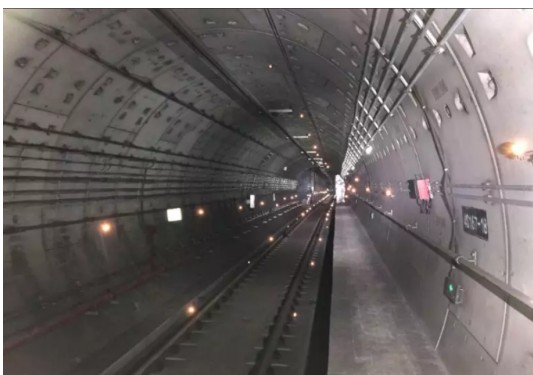

**Figure 2.** Photo of the measuring instruments used for deformation data collection.

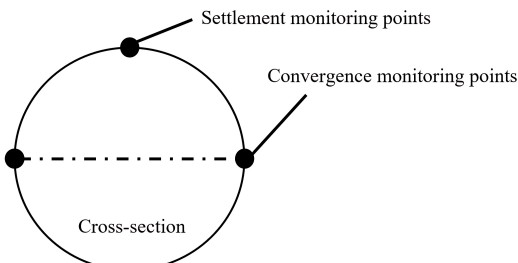

**Figure 3.** Schematic diagram of tunnel cross-section monitoring points layout.

This paper is based on the longitudinal settlement monitored data from 2010 to 2020 for the Xinjian Road tunnel and transverse convergence deformation monitored data from 2011 to 2020 for the Dalian Road tunnel. The two tunnels are divided into east and west lines for operation, with the core sections including the shield sections and rectangular sections. Figure 4 shows a schematic diagram of the monitoring points layout for the Xinjian Road tunnel. The east line, EXJ001A–EXJ039A, EXJ130A–EXJ141A, and the west line, WXJ001A–WXJ018A, WXJ101A–WXJ129A belong to the rectangular section, while the east line, EXJ040A–EXJ129A, and west line, WXJ019A–WXJ100A, belong to the shield section. Figure 5 shows a schematic diagram of the monitoring points layout for the Dalian Road tunnel. The east line, EDM1–EDM2, EDM9–EDM10, and west line, WDM1–WDM2, WDM9–WDM10, belong to the rectangular section, while the east line, EDM3–EDM8, and west line, WDM3–WDM8, belong to the shield section.

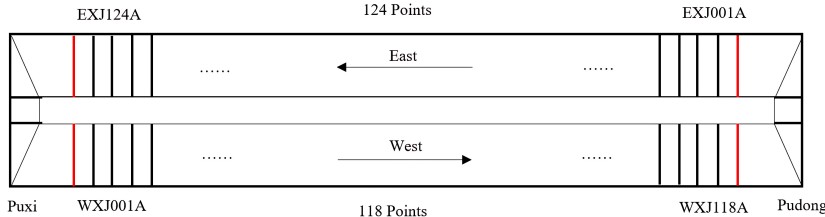

**Figure 4.** Layout of monitoring points on the east and west lines of the Xinjian Road tunnel.

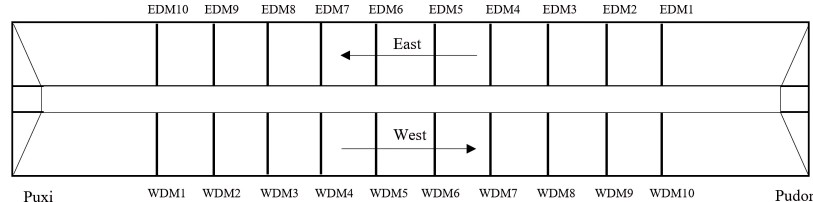

**Figure 5.** Layout of monitoring points on the east and west lines of the Dalian Road tunnel.

Table 1 presents the basic information on the structural deformation data for the two tunnels. Both sets of data were monitored 45 times and contain no missing or abnormal values.

**Table 1.** Experiment tunnel datasets.

| Tunnel Name | Opening Time | Measurement Time | Line | Points Number | Data Volume | Content |
|---|---|---|---|---|---|---|
| Xinjian Road | Mar. 2010 | Jan. 2010–Jun. 2020 | West | 118 | 5310 | Settlement |
| | Mar. 2010 | Jan. 2010–Jun. 2020 | East | 124 | 5580 | Settlement |
| Dalian Road | Sep. 2003 | Dec. 2011–Nov. 2020 | West | 10 | 370 | Convergence |
| | Sep. 2003 | Dec. 2011–Nov. 2020 | East | 10 | 370 | Convergence |

The original data used in this experiment are the elevation values of settlement measurement points and transverse length values, in meters. Given that the structural deformation of the tunnel mainly focuses on the cumulative changes in settlement and transverse deformation, it is essential to preprocess the raw data. To demonstrate, we select the first five data of point WXJ022A of the Xinjian Road tunnel. The conversion method is: the elevation measured at the monitoring point at a certain time minus the elevation of the point at the initial measurement time, and then multiplied by 1000 to convert the unit into millimeters. The specific data form is shown in Table 2.

**Table 2.** Monitored data of WXJ022A of the Xinjian Road tunnel (partial).

| Measurement Time | Elevation (m) | Cumulative Settlement (mm) |
|---|---|---|
| 1 Jan. 2010 | −18.90434 | 0.00000 |
| 1 Apr. 2010 | −18.90570 | −1.36000 |
| 1 Jul. 2010 | −18.90791 | −3.57000 |
| 1 Oct. 2010 | −18.90728 | −2.94000 |
| 1 Jan. 2011 | −18.90575 | −1.41000 |

The data on five timestamps of all the monitoring points of the tunnel are selected, and the overall cumulative settlement change of the tunnel is plotted, as shown in Figure 6 for the west line of the Xinjian Road tunnel. The overall cumulative convergence deformation

change of the tunnel is plotted, as shown in Figure 7 for the east line of the Dalian Road tunnel, for example. The accumulated deformation of both the shield and rectangular sections of the tunnel increases gradually with time, and the overall trend of the tunnel is upward due to the buoyancy effect. The settlement or convergence of the same section near the monitoring points behaves similarly, and the changes differ significantly at the connections of different sections.

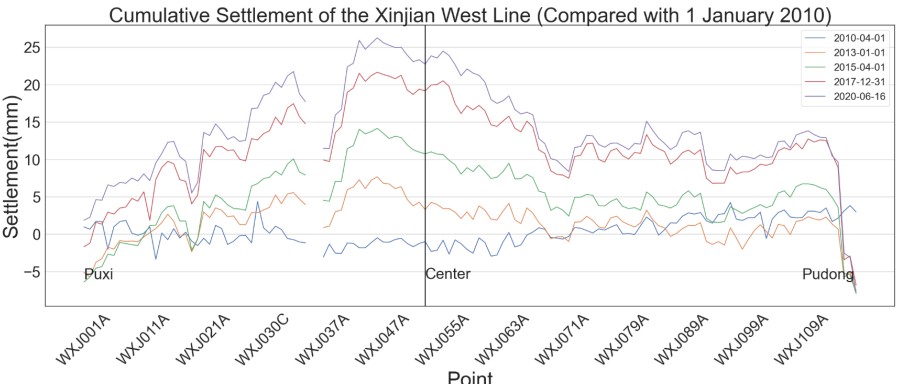

**Figure 6.** Cumulative settlement changes of the west line of the Xinjian Road tunnel.

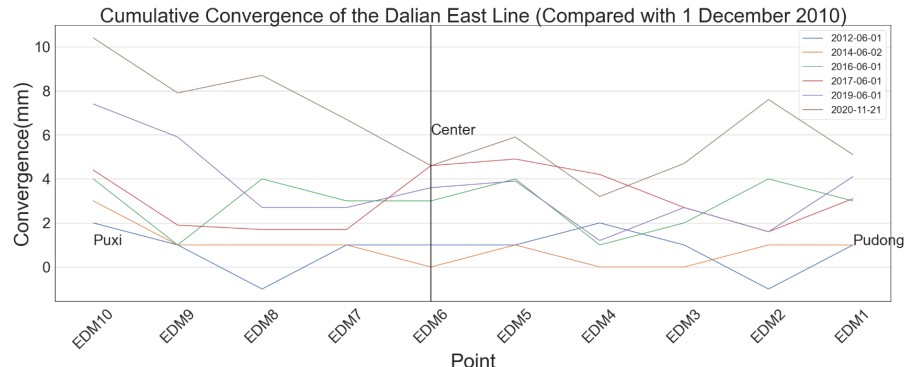

**Figure 7.** Cumulative convergence changes of the east line of the Dalian Road tunnel.

## 4. Model Design

### 4.1. Research Idea

Due to the clear time characteristics of deformation data collected from the tunnel, this paper defines the deformation of tunnel structures during operation as a time-series prediction problem, which uses historical deformation data to predict future tunnel deformation. Within the same tunnel structure, neighboring monitoring points may exhibit similarities in long-term deformation. However, most research focuses on individual monitoring points, without considering the correlation between monitoring points along different sections of the tunnel. This may limit the predictive capability of the model. Therefore, this paper models deformation at the level of the single monitoring point and the entire section of the tunnel, comparing their predictive performance. Commonly used models in the field of time-series prediction include grey models, linear regression models, and neural network models. Given the seasonal, irregular, and long-term trends exhibited in the data, and the single-input and multi-output structure, we select the traditional statistical method ARIMA and the deep learning method LSTM for comparison experiments.

After defining the research approach and experimenting with the methods, we further refine the experimental content. Firstly, we construct and process appropriate model inputs from the perspectives of individual monitoring points and sections of the tunnel. From the perspective of individual monitoring points, we follow the traditional prediction approach, using time as the only variable, and construct samples from time-series data using a fixed-length sliding time window to meet the requirements of supervised learning. Secondly,

we apply the ARIMA and LSTM models to each monitoring point, output the predicted values, and calculate model indicators by comparing them with real values. From the perspective of tunnel sections, we divide the core sections of the east and west tunnels into shield sections and rectangle sections, with the east section facing Pudong and the west section facing Puxi. As monitoring points within the same section of the tunnel are closely related, their time-series data changes showed similar patterns. By constructing samples from all monitoring points data within the same section of the tunnel, we are able to increase sample size and enhance input features. Applying the ARIMA and LSTM models to the entire section of the tunnel, we output the predicted average value for that section and calculate model indicators by comparing them with real values. Finally, we conduct multidimensional comparisons of the experimental results to enrich the results.

*4.2. Model Selection*

The ARIMA algorithm is a classical time-series prediction statistical method widely used in various fields. The Long- Short-Term Memory network LSTM is one of the most outstanding deep neural network algorithms in the prediction of time-series data. Therefore, this paper chooses these two models as the main body of the experiment.

4.2.1. ARIMA Model

The ARIMA model consists of three parts: Auto-Regressive model (AR), Moving Average model (MA), and difference method. Among them, AR is used to describe the relationship between the current value and the historical value, and the general *p*-order AR model is expressed as:

$$X_t = \alpha_1 X_{t-1} + \alpha_2 X_{t-2} + \ldots + \alpha_p X_{t-p} + u_t \tag{1}$$

where $X$ represents sequence data at different $t$ stages, $\alpha$ represents parameters, and $u_t$ represents random perturbation items. If $u_t$ is a white noise sequence, it is called a pure AR(p) process. If $u_t$ is not a white noise sequence, it is usually considered to be a moving average of order $q$, just as:

$$u_t = \varepsilon_t + \beta_1 \varepsilon_{t-1} + \ldots + \beta_q \varepsilon_{t-q} \tag{2}$$

where $\varepsilon_t$ represents a white noise sequence. When $X_t = u_t$, the MA model is obtained.

For the non-stationary time series, the *d*-order difference is first carried out to convert it into a stationary time series. Secondly, the Auto Correlation coefficient (ACF) and Partial Auto Correlation coefficient (PACF) are obtained for the stationary time series, along with the auto correlation graph and partial auto correlation from the analysis of the graph; the optimal order, *p*, *q*, is obtained. the ARIMA model is obtained from the *d*, *p*, *q* obtained above.

$$X_t = \alpha_1 X_{t-1} + \alpha_2 X_{t-2} + \ldots + \alpha_p X_{t-p} + \varepsilon_t + \beta_1 \varepsilon_{t-1} + \ldots + \beta_q \varepsilon_{t-q} \tag{3}$$

This paper conducts two types of experiments on datasets using ARIMA models. The modeling process includes preprocessing deformation data for each monitoring point and conducting tests for sequence stationarity and white noise. The order of differencing for each model is set to 1, and the optimal *p* and *q* values are determined based on the Bayesian Information Criterion (BIC), which meets the input requirements of ARIMA models. The dataset is then divided into training and testing sets. In order to demonstrate the multi-perspective predictive performance of the model better, this experiment separately models and predicts the input data from the perspectives of single monitoring points and sections of the tunnel. The input of the single point model is the time-series sample of each monitoring point and the input of the section model is the sample of all monitoring points within each section, with the output being a single value. The model is then trained, and the structure

and parameters are adjusted according to the prediction performance on the testing set. Finally, the model will be evaluated. The model process is shown in Figure 8.

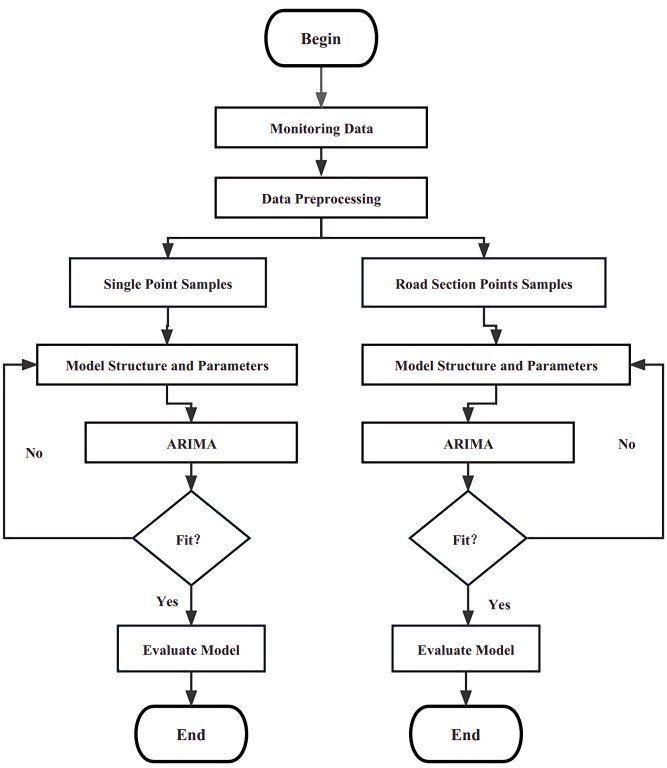

**Figure 8.** ARIMA modeling process.

### 4.2.2. LSTM Model

LSTM is a cyclic neural network suitable for long time-series data, which can reduce the problem of gradient vanishing and gradient explosion. Due to its time memory unit, LSTM can learn long short -term dependent information in time series, so it has a better performance in predicting long-term and time-series data with interval and delay. Figure 9 shows the structure diagram of the model elements of LSTM.

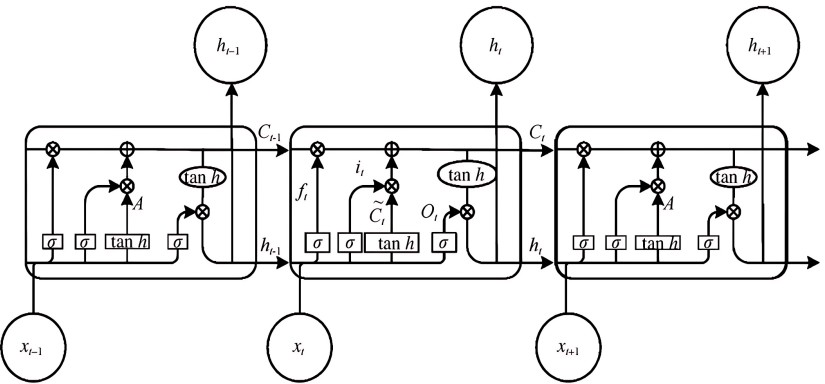

**Figure 9.** LSTM structure diagram.

The inputs of the LSTM model are: $x_t$ (input value at $t$), $h_{t-1}$ (output value at $t-1$), and $C_{t-1}$ (gate control unit state at $t-1$); The outputs are $h_t$ (output value at $t$) and $C_t$ (gate

control unit status at *t*). The following is the expression formula for forgetting gate, input gate, and output gate:

$$\begin{aligned} f_t &= \sigma(W_f h_{t-1} + W_f x_t + b_f), \\ i_t &= \sigma(W_i h_{t-1} + W_i x_t + b_i), \\ o_t &= \sigma(W_o h_{t-1} + W_o x_t + b_o) \end{aligned} \tag{4}$$

Among them, $f_t$, $i_t$, $o_t$ are the outputs of the forgetting gate, input gate, and output gate; $W_f$, $W_i$, $W_o$ are the weight matrices of forgetting gate, input gate, and output gate; $b_f$, $b_i$, $b_o$ are the bias terms for forgetting gate, input gate, and output gate; $\sigma$ is the activation function. By combining the output gate and unit status, the final output of LSTM can be determined using the following formula:

$$\begin{aligned} \tilde{C}_t &= \tanh(W_c h_{t-1} + W_c x_t + b_c), \\ C_t &= f_t C_{t-1} + i_t \tilde{C}, \\ h_t &= o_t \tanh(C_t) \end{aligned} \tag{5}$$

Based on LSTM, this paper needs to conduct experiments on two types of datasets. The modeling process is as follows. Firstly, preprocess the deformation monitored data for each monitoring point, and then divide the dataset into a training set and a test set. In order to better reflect the multi-angle prediction effect of the model, this experiment models and predicts the input data from the perspectives of a single point and a road section, where the input data for the single point model is the time-series sample for each point, and the input data for the road section model is the sample for all points on each road section. Secondly, determine the model structure and parameters. The number of neurons in the input layer is determined by the predicted input steps, and the number of neurons in the output layer is determined by the predicted output steps, with a single step output. Once the model structure is determined, set the initial parameters and iterations for model training. Adjust the model structure and parameters based on the prediction effect of the test set. Finally, evaluate the model. The model process is shown in Figure 10.

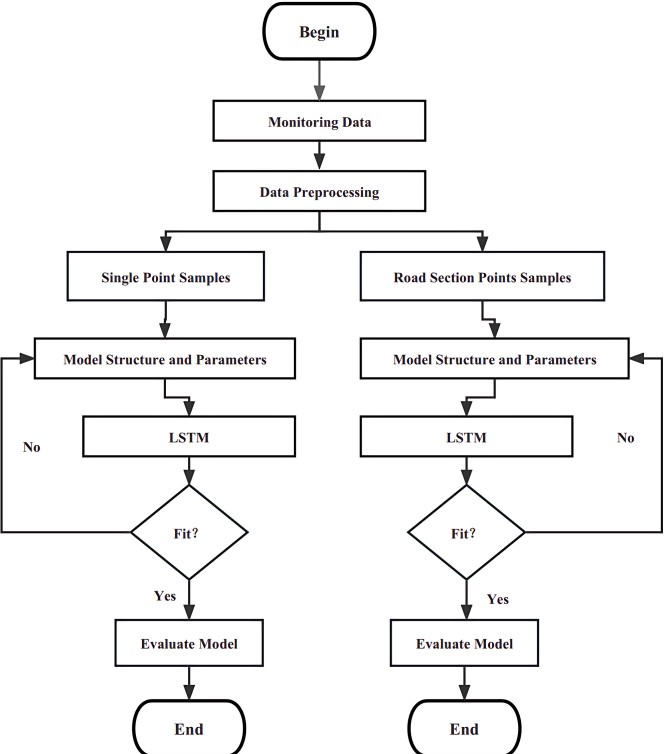

**Figure 10.** LSTM modeling process.

**5. Experiment Introduction**

*5.1. Experiment Design*

In this paper, there are four groups of experiments:

- Experiment 1: Settlement Prediction Based on Single Point;
- Experiment 2: Convergence Prediction Based on Single Point;
- Experiment 3: Settlement Prediction Based on Road Section;
- Experiment 4: Convergence Prediction Based on Road Section.

Each of these uses ARIMA and LSTM for prediction comparison; there are a total of eight models, and each model is given a number to distinguish it. Each model has multiple inputs and a single output. The experimental design and parameter settings are shown in Table 3 below. Based on longitudinal settlement and transverse convergence data, ARIMA and LSTM will be comprehensively compared and analyzed from the perspective of single monitoring points or road sections.

**Table 3.** Experiment scheme.

| Experiment | Model | Input Size | Train-Test Proportion | Parameters |
|---|---|---|---|---|
| Experiment 1 | ARIMA_S_S | 15 | 5:1 | p = 1, d = 1, q = 1 |
| | LSTM_S_S | 15 | 5:1 | Iter = 1500, lr = 0.0001, batch size = 100 |
| Experiment 2 | ARIMA_C_S | 12 | 4:1 | p = 1, d = 1, q = 0 |
| | LSTM_C_S | 12 | 4:1 | Iter = 1500, lr = 0.0001, batch size = 100 |
| Experiment 3 | ARIMA_S_R | 15 | 5:1 | p = 1, d = 1, q = 1 |
| | LSTM_S_R | 15 | 5:1 | Iter = 2000, lr = 0.0001, batch size = 100 |
| Experiment 4 | ARIMA_C_R | 12 | 4:1 | p = 1, d = 1, q = 0 |
| | LSTM_C_R | 12 | 4:1 | Iter = 2000, lr = 0.0001, batch size = 100 |

*5.2. Evaluation Indicators*

This paper evaluates the prediction model from three dimensions: accuracy, fitting degree, and explainability. Two types of indicators, MAE and MSE, belong to accuracy evaluation. MSE can reflect the influence of prediction anomalies, while MAE has a certain robustness to prediction anomalies. $R^2$ analyzes the fitting degree of the model, and Explained Variance is used for explainable analysis in the model, which is the degree to which the input variables of the model combine to affect the output.

Assume that for the time-series forecasting model, there are $n$ samples, each sample is $(x_i, y_i)$, and the predicted value is $\hat{y}_i$, $i \in 1, 2, \ldots n$. $\bar{y}$ is the mean value of $\{y_i\}_{i=1}^{n}$. The following are the meanings and formula definitions of each evaluation indicator.

5.2.1. MSE

MSE indicates the sum of squares of the differences between the predicted and true values. In general, the smaller the value of MSE, the better the model fits the data.

$$\text{MSE}(y, \hat{y}) = \frac{1}{n} \sum_{i=1}^{n} (y_i - \hat{y}_i)^2 \tag{6}$$

5.2.2. MAE

MAE indicates the absolute value of the difference between the predicted value and the true value. In general, the smaller the value of MAE, the better the model fits the data.

$$\text{MAE}(y, \hat{y}) = \frac{1}{n} \sum_{i=1}^{n} |y_i - \hat{y}_i| \tag{7}$$

### 5.2.3. $R^2$ (R-Squared)

$R^2$ (R-squared), also known as the coefficient of resolvability, reflects the degree to which the independent variable explains the changes in the dependent variable. The closer $R^2$ approaches 1, the better the model fits the data; the closer it approaches 0, the worse the model fits the data.

$$R^2(y, \hat{y}) = 1 - \frac{\sum\limits_{i=0}^{n} (y_i - \hat{y}_i)^2}{\sum\limits_{i=0}^{n} (y_i - \bar{y}_i)^2} \tag{8}$$

### 5.2.4. EV (Explained Variance)

EV is usually used to evaluate the explanation degree of a model for the fluctuations of the dataset. Its value is less than or equal to 1. Similar to $R^2$, the closer the EV value approaches 1, the more the model can completely explain the fluctuations of the dataset. On the other hand, the smaller the value, the worse the model's ability to explain the fluctuations of the dataset.

$$\text{ExplainedVariance}(y, \hat{y}) = 1 - \frac{\text{Var}\{y - \hat{y}\}}{\text{Var}\{y\}} \tag{9}$$

## 6. Results and Analysis

### 6.1. Comparison of Experiments Based on Single Point

#### 6.1.1. Based on Settlement Data

In Experiment 1, the ARIMA model and LSTM model are respectively established for the settlement data of each monitoring point on the east and west lines of the Xinjian Road from the perspective of a single point. The predicted values of each monitoring point are output, compared with the true values, and the evaluation indicators MAE, MSE, $R^2$, and EV are calculated. Table 4 shows the average number of evaluation indicators, better test points, and performance ratio of all test points on the east and west lines of the Xinjian Road in Experiment 1. The number of perfect points refers to the number of monitoring points with an MAE less than 2, MSE less than 5, and $R^2$ and EV greater than 0.75. The perfect proportion refers to the proportion of perfect monitoring points to the total monitoring points.

**Table 4.** Statistics of model indicators in Experiment 1.

| Model | Indicator | Mean | Perfect Number | Perfect Proportion |
|---|---|---|---|---|
| ARIMA_S_S | MAE | 1.8758 | 195 | 71.69% |
| | $R^2$ | 0.8155 | 210 | 76.00% |
| | EV | 0.8516 | 239 | 87.87% |
| | MSE | 4.3531 | 231 | 84.93% |
| LSTM_S_S | MAE | 2.1159 | 176 | 64.71% |
| | $R^2$ | 0.7653 | 178 | 65.44% |
| | EV | 0.8408 | 233 | 85.66% |
| | MSE | 5.5306 | 184 | 67.65% |

Figure 11 depicts the distribution of the values of the four indicators for all monitoring points under the two models. It can be seen that both ARIMA and LSTM show relatively excellent prediction performance, with a lower mean and median for MAE and MSE, indicating a smaller prediction error. $R^2$ and EV are both very close to 1, indicating a higher degree of model fitting. However, a comprehensive comparison reveals that ARIMA performs better than LSTM overall, showing a fairly high stability. ARIMA is smaller than LSTM in both MAE and MSE, while $R^2$ and EV are both higher than LSTM. It can also be

found in the box plot that ARIMA has a more uniform distribution of values with smaller standard deviation and fewer extreme values. This performance may be related to the overall more stable variation of the tunnel data. In contrast, the prediction performance of LSTM is relatively unstable, with a large standard deviation, and it is prone to outliers and extreme values. Although individual monitoring points perform better than ARIMA, the robustness is not strong.

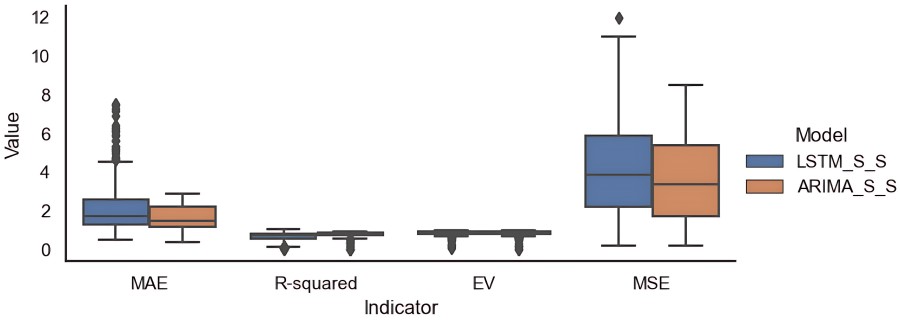

**Figure 11.** Value distribution of indicators in Experiment 1.

6.1.2. Based on Convergence Data

Experiment 2 also applies ARIMA and LSTM models to the Dalian Road transverse deformation dataset based on the single monitoring point perspective. Table 5 shows the output results of evaluation indicators for all monitoring sections on the east–west line of Dalian Road. Both LSTM and ARIMA models not only have stable prediction effect in settlement, but also have better performance in convergence, and the perfect proportion of both models in MAE and MSE are over 90%. The prediction error is further reduced.

**Table 5.** Statistics of model indicators in Experiment 2.

| Model | Indicator | Mean | Perfect Number | Perfect Proportion |
|---|---|---|---|---|
| ARIMA_C_S | MAE | 1.1312 | 19 | 95.00% |
| | $R^2$ | 0.7720 | 17 | 85.00% |
| | EV | 0.8271 | 18 | 90.00% |
| | MSE | 2.0984 | 19 | 95.00% |
| LSTM_C_S | MAE | 1.2181 | 18 | 90.00% |
| | $R^2$ | 0.7589 | 15 | 75.00% |
| | EV | 0.7791 | 16 | 80.00% |
| | MSE | 2.5140 | 18 | 90.00% |

For comparison, the median and the best performance of the results of Dalian Road are used as the benchmarks to construct Figure 12 for each evaluation indicator. From the two indicators, $R^2$ and EV, the robustness of LSTM is weak; its median is numerically lower than that of ARIMA. The prediction performance of different monitoring points varies widely, but the best performance in some monitoring sections is better than those of the ARIMA model.

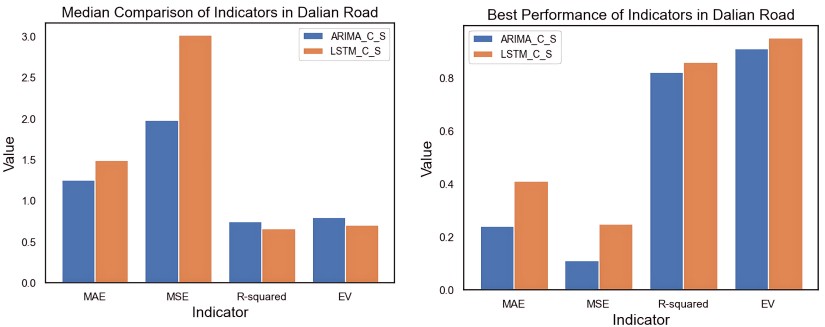

**Figure 12.** Comparison of Dalian Road model indicators in Experiment 2.

*6.2. Comparison of Experiments Based on Road Sections*

6.2.1. Based on Settlement Data

Experiment 3 divides the east and west tunnel sections of the Xinjian Road into west rectangular section, east rectangular section, west shield section, and east shield section according to the location and structure from the perspective of sub-sections. The experiment applies ARIMA and LSTM to all monitoring points within the same section to output the respective predicted values, compares them with the true values, and calculates the indicators. The indicators for each section are averaged to obtain the predicted performance of the section, and the results are shown in Table 6.

**Table 6.** Statistics of model indicators in Experiment 3.

| Model | Indicator | West | | | | East | | | | Mean |
|---|---|---|---|---|---|---|---|---|---|---|
| | | W-Rec | E-Rec | W-Shield | E-Shield | W-Rec | E-Rec | W-Shield | E-Shield | |
| ARIMA_S_R | MAE | 1.3260 | 1.4496 | 1.2740 | 1.5335 | 1.2330 | 1.2085 | 1.2103 | 1.5131 | 1.3435 |
| | $R^2$ | 0.8286 | 0.8161 | 0.8380 | 0.8316 | 0.8420 | 0.8368 | 0.8467 | 0.8286 | 0.8335 |
| | EV | 0.8692 | 0.8528 | 0.8879 | 0.8939 | 0.8780 | 0.8608 | 0.9095 | 0.8621 | 0.8767 |
| | MSE | 2.6975 | 3.1676 | 2.1113 | 2.0770 | 1.9440 | 1.8977 | 2.0543 | 1.2375 | 2.1483 |
| LSTM_S_R | MAE | 1.0760 | 1.1975 | 1.5348 | 1.4557 | 1.4298 | 1.6883 | 1.3567 | 1.2835 | 1.3777 |
| | $R^2$ | 0.8334 | 0.8293 | 0.8244 | 0.8231 | 0.8219 | 0.8133 | 0.8395 | 0.8218 | 0.8258 |
| | EV | 0.8915 | 0.8663 | 0.8561 | 0.8558 | 0.8547 | 0.8417 | 0.8953 | 0.8593 | 0.8650 |
| | MSE | 1.5069 | 1.8163 | 2.7026 | 2.9684 | 2.6644 | 3.4322 | 1.6870 | 2.9542 | 2.4665 |

From the comprehensive comparison of the western rectangular section, it can be seen that the relative growth rate of LSTM is higher than that of ARIMA. The average of $R^2$ in all sections of the western and eastern lines is more than 0.8, which means that the performance of LSTM on the section model has been greatly improved. For visual display and comparison, $R^2$ at different sections of the east and west lines of the Xinjian Road tunnel is selected for visualization to observe the performance of the model in different lines and sections. Figures 13 and 14 show the $R^2$ evaluation visualization of the model at the monitoring points of the west line and the east line of the Xinjian Road. Through observation and analysis, there are some differences in the performance of the models in different lines and different sections of the same tunnel. As for the Xinjian Road tunnel, the performance of the two models in the west line is better than that in the east line, but at the same time, the performance of the monitoring points where $R^2$ is located at the junction of the road section is poor. It is considered that the differential settlement at the junction may have an impact on the data quality due to different construction methods, soil structures, and other reasons, thus affecting the performance of the model. Although LSTM fluctuates greatly on different road sections, its $R^2$ best case occurs more frequently than ARIMA. If the data quality is enhanced, it may bring better results. ARIMA, on the other hand, remains stable and efficient and has strong generalization ability.

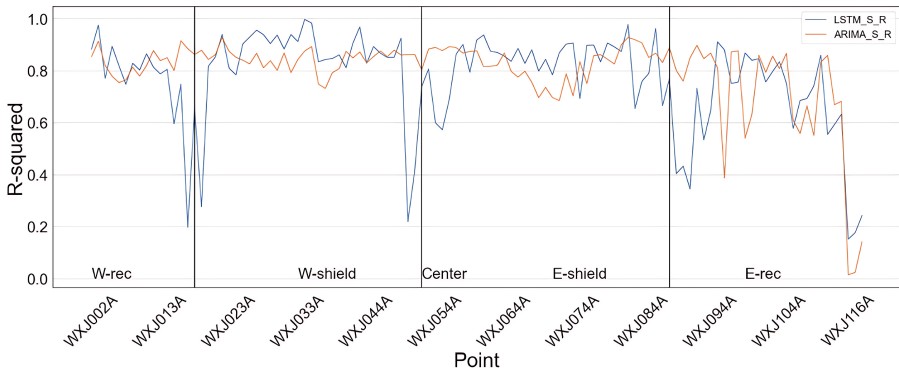

**Figure 13.** $R^2$ of the west line road section of Xinjian Road tunnel.

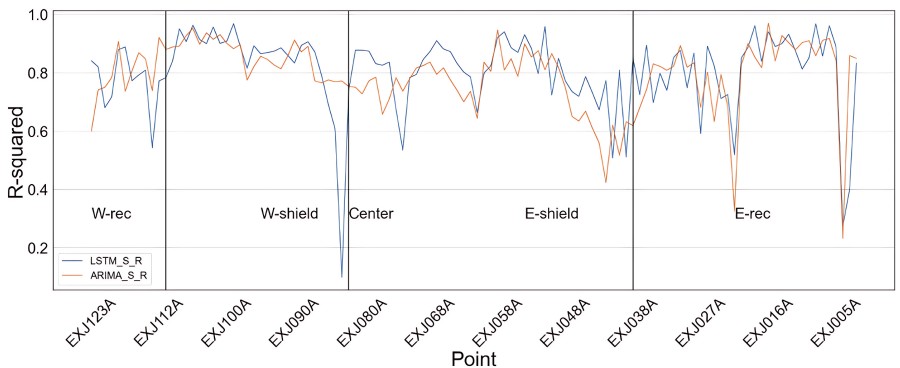

**Figure 14.** $R^2$ of the east line road section of Xinjian Road tunnel.

#### 6.2.2. Based on Convergence Data

In Experiment 4, ARIMA and LSTM models are also applied to the Dalian Road convergence dataset based on the road section angle. Table 7 is the statistical table of the indicators of the fourth model. Similar to the settlement prediction experiment of the Xinjian Road, in general, the indicators of the two models have been improved to different degrees.

**Table 7.** Statistics of model indicators in Experiment 4.

| Model | Indicator | West | | | | East | | | | Mean |
|---|---|---|---|---|---|---|---|---|---|---|
| | | W-Rec | E-Rec | W-Shield | E-Shield | W-Rec | E-Rec | W-Shield | E-Shield | |
| ARIMA_C_R | MAE | 0.9151 | 0.899 | 1.2166 | 1.1332 | 0.9757 | 1.0405 | 0.9462 | 0.9380 | 1.0080 |
| | $R^2$ | 0.8916 | 0.9032 | 0.8469 | 0.8646 | 0.8702 | 0.8791 | 0.8801 | 0.8827 | 0.8773 |
| | EV | 0.9159 | 0.9001 | 0.8749 | 0.8858 | 0.8982 | 0.8877 | 0.9019 | 0.9088 | 0.8966 |
| | MSE | 1.3875 | 1.1751 | 1.5079 | 1.2672 | 1.3716 | 1.4154 | 1.3716 | 1.3520 | 1.3560 |
| LSTM_C_R | MAE | 0.9404 | 0.9302 | 1.3692 | 1.0778 | 0.9016 | 1.1582 | 0.9482 | 0.9830 | 1.0386 |
| | $R^2$ | 0.8837 | 0.8791 | 0.8357 | 0.8539 | 0.8920 | 0.8470 | 0.8853 | 0.8708 | 0.8684 |
| | EV | 0.9034 | 0.8985 | 0.8410 | 0.8781 | 0.9035 | 0.8663 | 0.8907 | 0.8908 | 0.8840 |
| | MSE | 1.2399 | 1.1694 | 1.9688 | 1.6916 | 1.1968 | 1.9792 | 1.4760 | 1.7334 | 1.5568 |

As far as the $R^2$ visualization of the convergence of Dalian Road in Figures 15 and 16 is concerned, LSTM and ARIMA have similar performance in many sections of the east–west line due to the small number of sections. The reason may be that the geological environment or water level characteristics of the same section of the east line and the west line are similar. The impact on tunnel deformation is also similar. Therefore, the points in the same road section show consistent performance in the east and west line models.

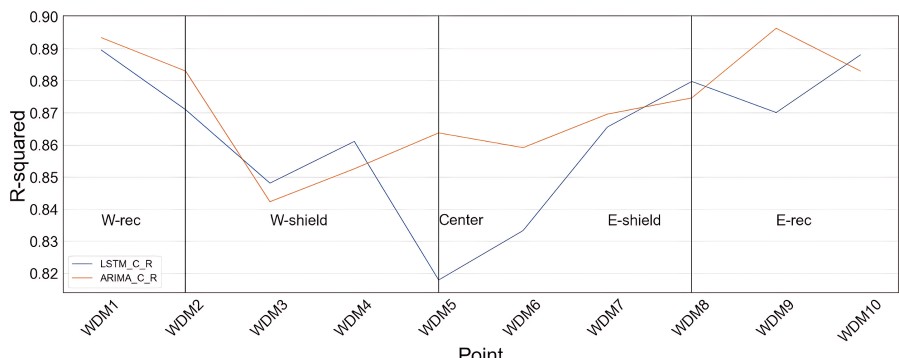

**Figure 15.** $R^2$ of west line road section of the Dalian Road tunnel.

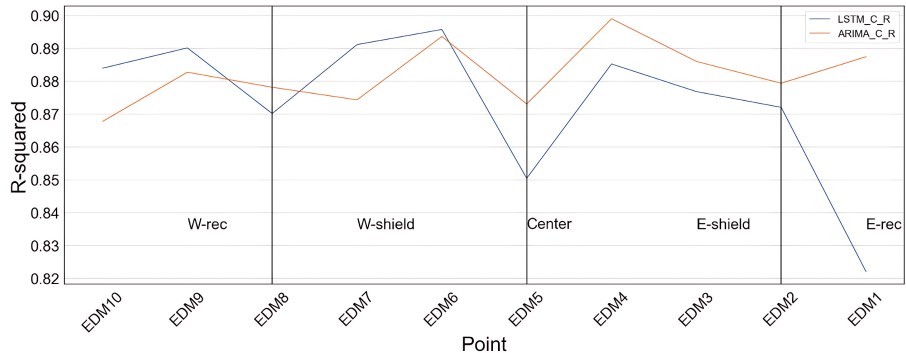

**Figure 16.** $R^2$ of east line road section of the Dalian Road tunnel.

Through sectional comparison and analysis, the prediction of settlement and convergence of the same model has many common points in the tunnel section, such as the difference of $R^2$ fitting in different sections and routes and the similarity of adjacent monitoring points. In addition, from the perspective of the overall fitting trend, ARIMA and LSTM tend to have similar performance in deformation prediction. This may be due to the fact that the information contained in the time-series data as a single variable cannot fully reflect the development law of deformation, leading to the similarity of the model after training, which needs to be supplemented and improved in future research.

*6.3. Comparison of Experimental Results between Models*

6.3.1. ARIMA

ARIMA single monitoring points were selected based on the same type of data to group and compare with the road model, and the results are shown in Table 8. It can be seen that for the settlement data, the average values of the indicators MAE and MSE of the ARIMA sub-section model have decreased significantly, with MAE decreasing by 28.38% and MSE decreasing by 50.65%, and the average values of $R^2$ and EV have increased slightly, with $R^2$ increasing by 2.21% and EV increasing by 2.95%. For the transverse convergence data, the average values of MAE and MSE of the model also decrease, with MAE decreasing by 10.90% and MSE decreasing by 35.38%, which is lower than the settlement data. However, $R^2$ has increased by 13.64% and EV has increased by 8.4%, which is larger than the settlement data. This indicates that the model makes it easier to fit deformation trends after enhancing the transverse convergence data samples.

**Table 8.** Comparison of the mean value of ARIMA model indicators.

| Data | Model | MAE | $R^2$ | EV | MSE |
|---|---|---|---|---|---|
| Settlement | ARIMA_S_S | 1.8758 | 0.8155 | 0.8516 | 4.3531 |
| | ARIMA_S_R | 1.3435 | 0.8335 | 0.8767 | 2.1483 |
| Convergence | ARIMA_C_S | 1.1312 | 0.7720 | 0.8271 | 2.0984 |
| | ARIMA_C_R | 1.0080 | 0.8773 | 0.8966 | 1.3560 |

6.3.2. LSTM

Observing the performance of LSTM in the road section and single monitoring point, the results of comparison are shown in Table 9. It can also be seen that the prediction performance of LSTM in the road section model is better than that of the single point model. Based on the settlement data, the average MAE value of the whole tunnel has decreased by 34.89%, the average $R^2$ value has increased by 7.91%, the average EV value has increased by 2.88%, and the average MSE value has decreased by 55.40%. Based on the transverse convergence data, the average MAE of the whole tunnel has decreased by 14.74%, the average $R^2$ has increased by 14.43%, the average EV has increased by 13.46%, and the average MSE has decreased by 38.07%.

**Table 9.** Comparison of the mean value of LSTM model indicators.

| Data | Model | MAE | $R^2$ | EV | MSE |
|---|---|---|---|---|---|
| Settlement | LSTM_S_S | 2.1159 | 0.7653 | 0.8408 | 5.5306 |
| | LSTM_S_R | 1.3777 | 0.8258 | 0.8650 | 2.4665 |
| Convergence | LSTM_C_S | 1.2181 | 0.7589 | 0.7791 | 2.5140 |
| | LSTM_C_R | 1.0386 | 0.8684 | 0.8840 | 1.5568 |

In a comprehensive comparison, ARIMA's indicators change more smoothly and LSTM's overall indicators change more significantly than ARIMA's. The mean value of MAE and MSE decreases less than ARIMA's, and the mean value of $R^2$ and EV increases more than ARIMA's. This shows that the prediction performance of LSTM has been greatly improved, compared with ARIMA in the road section model. The results may be due to the sufficient number of samples after segmentation, which improves the dataset quality. The learning of neural network is more sufficient than the traditional time-series model. LSTM can learn more information, and the model indicators improve faster.

**7. Summary and Prospect**

Focusing on predicting trends in structural deformation during the operation period of shield tunnels, this paper takes longitudinal uneven settlement deformation and transverse convergence deformation as experimental objects. Based on readily available time-series data, a comprehensive comparison and analysis of ARIMA and LSTM in data prediction performance is conducted from the point of view of single monitoring points and road sections. The experiments show that the ARIMA and LSTM models constructed in this paper both have good generalization performance and fitting ability, and can effectively and accurately predict deformation in most tunnel monitoring points. Overall, ARIMA has better stability and universality for small sample datasets, while the upper limit of LSTM is strong but slightly unstable. With higher quality data, model performance may be greatly improved.

The experiments also have practical implications for engineering. For important monitoring points in tunnels, the ARIMA or LSTM model with better performance corresponding to each point can be selected to predict deformation for a period of time in the future; record the predicted value; and establish a preventive alarm mechanism focusing on monitoring points that may have uplift, large settlement amplitude, obvious settlement difference with surrounding monitoring points, or that exceed the settlement warning

value in the future. Strengthening monitoring and management of the road sections where points are located or carrying out preventive maintenance can prevent future problems.

In future research, we need to further optimize and improve the dataset and experimental models used in the article. The following are the directions for improvement considered in this article:

1. At present, the model adopts a rolling prediction method, which is based on temporal information for deformation prediction. Although long-term dependencies on temporal data can be learned using ARIMA or LSTM, this dependency relationship is still not presented in the form of parameters. The interpretability of the model is not clear enough. The future improvement direction can be based on probability statistical methods to make long-term probability predictions of tunnel deformation, indicating whether the extension of the operation period has increased the probability of deformation occurrence.

2. The premise for establishing the model in this article is that the tunnel is in a long-term stable operating environment, but this assumption is not very appropriate. Although it is difficult to comprehensively consider the factors that affect tunnel deformation, it is beneficial to improve the accuracy of model prediction. In the future improvement direction, we will add the screening of factors, using popular methods such as random forest or XGBoost, and then include the factors with higher importance into the model.

3. At present, this article only conducts experiments on two tunnel datasets, the Xinjian Road and Dalian Road Tunnels. The universality and generalization ability of the model still need to be further verified through other datasets.

**Author Contributions:** Conceptualization, M.H.; methodology, M.H., C.D. and H.Z.; validation, M.H. and C.D.; formal analysis, M.H., C.D. and H.Z.; investigation, M.H. and H.Z.; data collection, M.H., C.D. and H.Z.; writing—original draft, M.H. and H.Z.; writing—review and editing, M.H. and C.D. All authors have read and agreed to the published version of the manuscript.

**Funding:** This research received no external funding.

**Data Availability Statement:** The data is confidential.

**Conflicts of Interest:** The authors declare no conflict of interest.

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
