# Peer review of "Comparison of ARIMA and LSTM in Predicting Structural Deformation of Tunnels during Operation Period"

_data, 2023_

Round 1

Reviewer 1 Report

- ARIMA and LSTM abbreviations should be explained in short in the abstract as those terms will not be known to all readers - they are explained only half through the paper.

- More references relevant to probabilistic assessment of tunnelling-related deformations should be provided.

- Line 22 - "diseases" is a term used for living things, here "problems" would be more appropriate.

Reviewer 2 Report

This study compares the performance of two models, ARIMA and LSTM, in predicting the structural deformation of tunnels during operation periods. The study uses historical data from two tunnels in Shanghai and proposes a new way of modeling based on single points and road sections. The results show that both models have great performance for settlement and convergence deformation, but ARIMA has better overall robustness and adaptability to datasets. The article emphasizes the importance of predicting tunnel structural deformation to ensure safety and prevent accidents.

Based on the detailed analysis of the content, here are some suggestions for improving the article:

1.       The article mentions that some previous models did not consider factors such as multi-lane usage, heavy load, and soil rheology, and their prediction results were smaller than actual conditions. It would be beneficial if the authors could clarify whether their proposed models have considered these factors and how they have addressed these limitations.

2.       The article mentions that the screening of correlated factors in some models was subjective, resulting in poor model generalization performance. It would be beneficial if the authors could explain how their proposed models overcome this limitation.

3.       The article mentions that some models are not suitable for long-term time series forecasting as the prediction error gradually increases with time. It would be beneficial if the authors could clarify whether their proposed models are suitable for long-term predictions and how they have addressed this limitation.

4.       The article mentions that the data used in the research is confidential. This could limit the reproducibility of the research and make it difficult for other researchers to validate the results.

5.       The article mentions various figures and diagrams, but without the actual images, it's hard to understand the points being made. Including clear, high-quality images and ensuring they are properly referenced in the text will greatly improve the reader's understanding.

6.       There are some sentences in the article that are quite long and complex. Breaking these down into shorter, simpler sentences could improve readability. Additionally, proofreading the article for any grammar or spelling mistakes would also be beneficial.

7.       The article could benefit from a section discussing the limitations of the current study and potential future work. This would give readers a better understanding of the scope of the research and where it could be headed in the future.

Extensive editing of the English language is required.

Round 2

Reviewer 2 Report

The authors have responded to most of the Points.

Minor editing of English language required.